# PCSK9 Plasma Levels Are Associated with Mechanical Vascular Impairment in Familial Hypercholesterolemia Subjects without a History of Atherosclerotic Cardiovascular Disease: Results of Six-Month Add-On PCSK9 Inhibitor Therapy

**DOI:** 10.3390/biom12040562

**Published:** 2022-04-09

**Authors:** Arianna Toscano, Maria Cinquegrani, Michele Scuruchi, Antonino Di Pino, Salvatore Piro, Viviana Ferrara, Carmela Morace, Alberto Lo Gullo, Egidio Imbalzano, Francesco Purrello, Giovanni Squadrito, Roberto Scicali, Giuseppe Mandraffino

**Affiliations:** 1Internal Medicine Unit, Department of Clinical and Experimental Medicine Lipid Center, University of Messina, 98122 Messina, Italy; arianna.toscano13@gmail.com (A.T.); mariacinquegrani@gmail.com (M.C.); mscuruchi@unime.it (M.S.); cmorace@unime.it (C.M.); eimbalzano@unime.it (E.I.); gsquadrito@unime.it (G.S.); gmandraffino@unime.it (G.M.); 2Department of Clinical and Experimental Medicine, University of Catania, 95124 Catania, Italy; nino_dipino@hotmail.com (A.D.P.); spiro@unict.it (S.P.); vivi.fer@hotmail.it (V.F.); francesco.purrello@unict.it (F.P.); 3Unit of Rheumatology, Department of Medicine, ARNAS Garibaldi Hospital, 95122 Catania, Italy; albertologullo@virgilio.it

**Keywords:** PCSK9, familial hypercholesterolemia, atherosclerotic injury

## Abstract

Proprotein convertase subtilisin/kexin type-9 (PCSK9) is a key regulator of low-density lipoprotein (LDL) metabolism involved in the degradation of the low-density lipoprotein receptor (LDLR) through complex mechanisms. The PCSK9 plasma levels change according to lipid lowering therapy (LLT). Few data exist regarding the role of PCSK9 in vascular damage. We aimed to evaluate the impact of PCSK9 plasma levels on pulse wave velocity (PWV) and the effect of PCSK9 inhibitors (PCSK9-i) on circulating PCSK9 and PWV in a cohort of heterozygous familial hypercholesterolemia (HeFH) subjects. In a previous step, HeFH patients were enrolled and LLT was prescribed according to guidelines. Biochemical analyses and PWV assessment were performed at baseline (T0), after 6 months of high-efficacy statin plus ezetimibe (T1) and after 6 months of PCSK9-i (T2). The PCSK9 levels were evaluated in 26 selected HeFH subjects at the three time points and 26 healthy subjects served as controls for the reference value for PCSK9 plasma levels. The PWV values decreased at each time point in HeFH subjects after LLT starting (8.61 ± 2.4 m/s, −8.7%; *p* < 0.001 vs. baseline at T1, and 7.9 ± 2.1 m/s, −9.3%; *p* < 0.001 vs. both T1 and baseline) and it was correlated to PCSK9 (r = 0.411, *p* = 0.03). The PCSK9 levels increased on statin/EZE therapy (+42.8% at T1) while it decreased after PCSK9-i was started (−34.4% at T2). We noted a significant relationship between PCSK9 levels and PWV changes at T1 and T2. In conclusion, PCSK9 levels were associated with baseline PWV values in HeFH subjects; moreover, we found that PCSK9 level variations seemed to be correlated with PWV changes on LLT. A longer observation time and wider sample size are needed to assess the potential role of PCSK9 plasma levels on the vascular function and remodelling, and to clarify the effects of PCSK9-i in these pathways.

## 1. Introduction

Atherosclerosis is the primary cause of worldwide cardiovascular disease morbidity and mortality [1]. Among several metabolic and environmental mechanisms, increased low-density lipoprotein cholesterol (LDL-C) level is considered the causal factor of atherosclerotic cardiovascular disease (ASCVD) [2]. Monogenic familial hypercholesterolemia is an autosomal dominant genetic condition characterized by elevated LDL-C levels from childhood; firstly, the majority of familial hypercholesterolemia (FH) mutations were found in the genes encoding for the low-density lipoprotein receptor (LDLR) and for apolipoprotein B. However, in 2003 the first FH related genetic variant was found in the gene encoding for proprotein convertase subtilisin/kexin type-9 (PCSK9) and an increasing attention has been drawn to the role of PCSK9 since then [3].

The serine protease PCSK9 is synthesized primarily by the liver and intestine as a 692-amino acid precursor (~75 kDa) and it undergoes several catalytic steps to reach its mature form [4]. Although its biological significance is not precisely known, the major function of PCSK9 seems to be the LDLR degradation [3].

Briefly, circulating PCSK9 binds to the LDLR and this complex is subsequently internalized in the hepatocytes. The binding of PCSK9 to the LDLR induces a modification of the LDLR conformation that enhances its degradation in the lysosome [4]. Thus, the density of the LDLR on the hepatocyte surface is inversely proportional to the PCSK9 plasma levels [5].

Gain of function (GOF) PCSK9 mutations were identified as the third genetic cause of autosomal dominant FH [5] and an increasing attention has been drawn to targeting PCSK9 for the reduction of LDL-C levels and the prevention of cholesterol-driven cardiovascular disease [6].

The effect of statin therapy on circulating PCSK9 concentrations has been previously studied [7,8]. High intensity statin treatment was associated with a significant increase in PCSK9 levels, both with atorvastatin [9,10,11,12,13]—likely following a dose-response effect [13]—and with rosuvastatin, simvastatin, pitavastatin or pravastatin treatment [14,15,16,17]. A recent meta-analysis by Sahebkar et al. confirmed these results and suggested that treatment with lipophilic statins such as atorvastatin, simvastatin, and pitavastatin lead to a greater increase in circulating PCKS9 levels than treatment with hydrophilic statins such as rosuvastatin and pravastatin [18].

The role of ezetimibe in relation to PCKS9 plasma levels was previously studied. In general, ezetimibe treatment seemed to increase circulating PCSK9 levels [19,20], although this increase could be no longer significant in addition to statin therapy [18,21]. The weaker LDL-C lowering effect of ezetimibe and exposure could explain this concept [22,23].

As concerns the lipid lowering pharmacological strategy, high-intensity statin is the cornerstone of lipid lowering therapy (LLT) in FH subjects and the addition of the PCSK9 inhibitor (PCSK9-i) effectively reduces the LDL-C amount; thus, the administration of PCSK9-i is needed especially in subjects at higher cardiovascular risk as FH [24]. The inhibition of PCSK9 prevents degradation of the LDLR and thus it increases its recycle on the hepatocyte surface with a subsequent reduction of circulating LDL-C. In all alirocumab studies, after 4 weeks of treatment an increase of the drug concentrations with a concomitant reduction of free PCSK9 concentrations was reported. The LDL-C changes from the baseline to week 24 after alirocumab starting were significant in all the trials and the reductions of free PCSK9 and LDL-C were also maintained.

In large cohort studies, the PCSK9 plasma levels have been shown to decrease after 4 weeks of PCSK9-i treatment and it continue to remain low if PCSK9-i administration was periodically repeated [25]. Furthermore, it was shown that circulating PCSK9 was less marked in subjects also assuming statin therapy [25]. The same finding was reported when evolocumab was used [26]; however, while unbound PCSK9 plasma levels are substantially undetectable when PCSK9-i plasma levels are the highest, a ten-fold increase of total PCSK9 levels was reported [26]. However, the clinical consequence of this change is currently not clarified.

In the last few years, scientific research has also focused on the concept of atherosclerosis as a chronic inflammatory disease [27]; in particular, inflammation has been considered the pathological expression of hypercholesterolemia and immune system dysfunction [28]. This concept could partially explain why, despite changes in lifestyle and the use of traditional lipid lowering therapy (LLT) to reduce plasma cholesterol levels, ASCVD is still the leading cause of death and loss of disability-adjusted life years [29]. The PCSK9 could play a role in this setting. Indeed, it has recently been suggested that PCSK9 could promote atherosclerosis progression by stimulating proinflammatory cytokine production and promoting oxidative stress within the atherosclerotic lesions, independently of the LDL-C levels [30].

In line with this hypothesis, a recent study showed that PCSK9-i treatment was able to reduce the LDL-C levels as well as the pulse wave velocity (PWV) in FH subjects [31]. The PWV measurement is a non-invasive and accurate instrumental method to estimate the arterial stiffness (AS) in clinical practice. Determinants of PWV, and reference values—also in relation to age and blood pressure values—have been established in a multicentric study by the Reference Values for Arterial Stiffness Collaboration [32]. In the last decade, additional evidence has emerged about its potential role as a marker of the mechanical vascular impairment and it was previously shown that PWV was associated with coronary, cerebral, and carotid atherosclerosis [33]. In particular, in 2017, Kubozono et al. first showed that a high PWV was a strong predictor of intimal media thickness (IMT) increase [34]. Furthermore, a recent longitudinal study by Yang et al. showed a correlation between the risk of carotid plaque formation and PWV, independently of other risk factors [35]. Finally, Zureik et al. showed a significant correlation between PWV and carotid plaque [36]. To the best of our knowledge, no data have been reported on the relationships between circulating PCSK9 levels and PWV changes in FH subjects on PCSK9-i administration.

In this study we aimed to evaluate the impact of PCSK9 plasma levels on mechanical vascular impairment evaluated by PWV and the effect of PCSK9-i on circulating PCSK9 and PWV in a cohort of FH subjects.

## 2. Materials and Methods

Subjects included in this study were randomly selected from a larger database, already collected for another recent study [31,37]. All participants were enrolled from the Lipid Centers of the University Hospital of Messina and the University Hospital of Catania, Italy, from September 2017 to May 2019; these are two tertiary centres for the screening, diagnosis and management of familial dyslipidemias. All participants were free of hematopoietic disorders, kidney injury, liver disease, malignancies and/or treatment with chemotherapy, acute infections, chronic inflammatory status, and glucocorticoid therapy within the past three months. Subjects with arterial hypertension, diabetes mellitus and ASCVD were also excluded. Genetic analysis of FH was carried out within the Lipigen program [38]. In total, 127 subjects were evaluated; 45 subjects had been classified as cardiovascular (CV) primary prevention patients; out of these, 19 patients were then excluded due to comorbidities, missing data, or missing stored samples. So, after checking that all needed samples (T0, T1, T2), as well as all the clinical and instrumental data, were available, 26 HeFH subjects satisfying the inclusion criteria were selected for this study, and PCSK9 levels were determined in 26 series of selected samples. As previously reported, for that study a blood sample was drawn and stored (processed within 2 h and frozen at −80 °C) at each time point. T0 was the baseline, T1 the time after 6 months of high-efficacy statin plus ezetimibe treatment and T2 the time after 6 months of iPCSK9 add-on therapy. The T2 blood withdrawal was scheduled for the 25th week to limit the potential variability of mAbs plasma levels. Moreover, 26 healthy controls were selected by SPSS case-control matching function to obtain an age- and sex-matched control population. In these subjects, the lipid profile and the PCSK9 levels were measured at the time of enrolment; blood pressure was also recorded.

At the time of enrolment, a physical examination was made and the clinical history of both HeFH subjects and healthy controls were recorded. All participants had standard haematological and clinical biochemistry parameters measured after a 12-h fast by standard laboratory tests. Body weight and height were measured, and the body-mass index was calculated as weight divided by the squared value of height (kg/m^2^). For the wider study, we evaluated complete lipid profile (including Apolipoprotein B (Apo-B), Apo-A, Lp(a)), liver transaminases, fasting plasma glucose, plasma insulin and inflammatory markers (see [31] for details). Overall, 78 frozen aliquots were tested, 26 for each of the three examination times for this observation. Twenty-six healthy subjects aged-matched to the participants served as controls, and to assess the reference value for PCSK9 plasma levels.

The study was approved by the local ethics committee (prot. Number 46/19).

### 2.1. Pulse Wave Velocity Evaluation

The SphygmoCor CVMS (AtCor Medical, Sydney, Australia) system was used for the determination of the PWV. This system uses a tonometer and two different pressure waves obtained at the common carotid artery (proximal recording site) and at the femoral artery (distal recording site). The distance between the recording sites and the suprasternal notch was measured using a tape measure. An electrocardiogram was used to determine the start of the pulse wave. The PWV was determined as the difference in interval time of the pulse wave between the two different recording sites and the suprasternal notch, divided by the travel distance of the pulse waveform. The higher the speed of travel, the higher the arterial stiffness; thus, a reduced PWV reflects an improved arterial elasticity. The PWV was calculated on the mean of 10 consecutive pressure waveforms to cover a complete respiratory cycle. We did not use prefixed cut-offs to classify normal or abnormal PWV. We considered PWV values compared to the control mean.

### 2.2. PCSK9 Plasma Levels

The PCSK9 levels were blindly measured using ELISA (Abcam, Human PCSK9 ELISA kit) with plasma aliquots collected after overnight fasting and stored at −80 °C within 2 h from blood collection. The minimum detectable dose ranged from 0.5 to 32 ng/mL. The PCSK9 levels are presented as both absolute values and n-fold difference with respect to the controls median.

### 2.3. Statistical Analysis

As verified by a Kolmogorov–Smirnov test, some variables selected for this study were non-normally distributed; therefore, a classic non-parametric approach was chosen. Consistently, data are presented as median (interquartile range), or as number and percentage where appropriate. The difference among the different timepoints was tested by Kruskal–Wallis test, and the Wilcoxon test was carried out to verify the difference between T1 and T0, T2 and T1, and between T2 and T0 as regards PCSK9 plasma levels, PWV values, and lipid parameters. The Δ change was calculated (following the formula: (N2 − N1)/N1%). The Mann–Whitney U test was used to verify the statistical difference between patients and controls as regards the study variables. A Spearman’s test was carried out to assess the interrelationships between the study variables. The interrelationships were then verified by univariate regression models where a pathobiological plausibility existed. Statistical analyses were performed using SPSS version 26 software (IBM SPSS ver. 26.0 64-bit, IBM corp. 2019). The SPSS case-control matching function was used to select control subjects from our healthy subjects’ database. A *p* < 0.05 was chosen to denote statistical difference.

## 3. Results

Table 1 summarizes the general characteristics of the 26 randomly selected HeFH patients and controls; plasma lipids were reported for each of the three time points, as well as the PCSK9 plasma levels, PWV values, SBP, and DBP. We noted that the HeFH patients exhibited a higher BMI than the controls: 27.1 (5.2) vs. 23.8 (1.8), *p* = 0.05; this finding was further analysed, as the PCSK9 plasma levels were also found to be increased in obese and T2DM young women [39], and the association was excluded in this study population by a univariate regression analysis. During the observation time, no patients experienced acute cardiovascular or cerebrovascular disease, nor were they diagnosed with type 2 diabetes mellitus (T2DM) or arterial hypertension.

In summary, we found a 46.3% average decrease in LDL-C after the prescription of high-efficacy statin-plus-ezetimibe LLT; as we had already found in the wider study, an additional 57.2% average decrease was obtained after six months of PCSK9-i added on therapy (Table 2, Figure 1A). We also found an 8.7% average decrease in PWV values at T1, and 9.3% average decrease at T2.

As regards statin therapy, all selected patients were on high-intensity statin regimen with rosuvastatin 20 mg (69.2%) or atorvastatin 40 mg (30.8%). All patients were also prescribed to assume ezetimibe 10 mg per day.

At the second time point, a monoclonal antibody inhibiting PCSK9 was started; more specifically, in 50% (13 patients), evolocumab 140 mg was prescribed, in 46.7% (12 patients), alirocumab 150 mg was prescribed, and in 3.33% (1 patient) alirocumab 75 mg was prescribed.

At baseline, average PCSK9 plasma levels were significantly higher than in control subjects (1.22 *n*-fold, *p* < 0.001, vs. baseline controls). After six months of therapy, the PCSK9 levels were significantly increased (1.77 *n*-fold, *p* < 0.001 both vs. the baseline controls and T0 patients). At T2, the levels were significantly lowered (1.17 *n*-fold, *p* < 0.001 vs. T1 patients), although slightly but significantly higher than baseline controls (*p* = 0.012) (Table 2, Figure 1B and Figure 2).

As concerns the PWV assessment, baseline PWV values were significantly higher in FH subjects than controls (9.6 ± 3.1 vs. 5.3 ± 0.65 m/s; *p* < 0.001). At T1, after initiation of high efficacy LLT with statins and ezetimibe, an 8.7% decrease in PWV was observed (8.61 ± 2.4 m/s, *p* < 0.001 vs. baseline). An additional 9.3% decrease was observed at T2, six months after iPCSK9 treatment was started (7.9 ± 2.1 m/s, *p* < 0.001 vs. T1); at T2, PWV values were still significantly higher than controls (*p* < 0.001) (Table 2, Figure 1C).

The potential association between plasma PCSK9 levels and PWV was verified; at baseline, the PCSK9 levels correlated with PWV values (r: 0.409, *p* = 0.025) as well as ΔPCSK9 with ΔPWV at T1 and T2, (*p* < 0.001). These interrelationships were then verified by univariate regression models, and we found that PWV decrease (ΔPWV) was associated with PCSK9 change in levels (ΔPCSK9) also at T1, when their median plasma levels increased; more specifically, the model tested suggested that the lower the PCSK9 increase at T1, the higher the PWV decrease (β = 0.690, t 2.863, *p* = 0.019). Consistently, we found the same relationship also at T2, where the higher PCSK9 decrease was associated with the higher PWV decrease (β = 0.602, t = 2.717, *p* = 0.018). No other variables were associated with PCSK9 plasma levels.

Of note, we found a slight and non-significant change in HDL-C levels over time. In detail, at T1 we found 1 mg/dL increase in the median value (53 mg/dL IQR 12 vs. 52 mg/dL IQR 11; *p* = 0.75) while at T2 a 4 mg/dL decrease compared to the baseline (48 mg/dL IQR 14 vs. 53 mg/dL IQR 12; *p* = 0.35). Few data exist about HDL-C changes on high intensity LLT including PCSK9-i. Most of the knowledge about HDL-C behaviour comes from studies on rosuvastatin; however, the evidence is conflicting with some studies showing an increase in HDL-C plasma levels, while others showed no change or little decrease [40]. Concerning PCSK9-i, most trials showed a little increase in HDL-C after PCSK9-i therapy. However, it has been suggested that the PCSK9 plasma levels could also be associated with HDL-C levels, and that PCSK9 lowering by its inhibitor could also be associated with HDL-C decrease [41], although this little decrease could not reflect a lower efficacy of HDL [40,41]. Moreover, the changes recorded in our study are not statistically significant.

## 4. Discussion

In the last few years, scientific research has focused on better evaluating the relationship between LDL-C and atherosclerotic cardiovascular disease. Consensus was reached about the importance of lowering LDL-C in patients at high and very high cardiovascular risk, as LDL-C was identified as a key player in the formation and progression of atherosclerotic disease. The “the lower the better” approach is supported by the most recent ESC/EAS Guidelines on Dyslipidemias, where the need to achieve more stringent LDL-C targets in patients at high and very high cardiovascular risk, such as patients with FH, is strongly emphasized [42]. The open question remains whether other factors could play an important role in atherosclerotic disease development. In the era of precision medicine, increasing attention has been drawn to the role of novel circulating biomarkers as pathological players of the atherosclerotic injury, especially in subjects at high cardiovascular risk. The PCSK9 appears to be a promising biomarker in the game of atherosclerosis; in fact, the PCSK9 plasma levels have been suggested to be linked with atherosclerosis progression by lipid and non-lipid pathways. As concerns the lipid pathway, Guardiola et al. previously found that high PCSK9 plasma levels were correlated with the atherogenic lipoprotein subclasses that are the main players of the foam cell in the atherosclerotic lesion [43]. As regards the non-lipid pathway, Ricci et al. found that PCSK9 plasma levels induced a pronounced pro-inflammatory state in circulating macrophages by promoting the stimulation of a set of chemokines and cytokines, thus enhancing stimulation and infiltration of monocytes in the arterial wall [44]. Taking these findings together, it may be hypothesized that circulating PCSK9 could be a dangerous enemy of cardiovascular health and its reduction could be beneficial for slowing atherosclerosis progression.

To evaluate the relationship between circulating PCSK9 and mechanical vascular impairment, we examined the correlation between baseline PCSK9 plasma levels and baseline PWV, we then compared these values with the PCSK9 plasma levels and PWV of healthy normolipidemic controls. A positive correlation was found between baseline PCSK9 and PWV in HeFH patients. The same was found to be true in healthy normolipidemic subjects. Finally, baseline PCSK9 plasma levels and PWV values were found to be higher in subjects at high cardiovascular risk with respect to controls. This was in line with a previous study by Ruscica et al. who demonstrated a positive relationship between circulating PCSK9 and PWV in a large cohort of subjects independently of LDL-C [45]. Hence, a reduction of PCSK9 plasma levels could be beneficial to improve mechanical properties.

Of note, previous findings suggested a role for PCSK9 in the vascular damage progression. In the *ATHEROREMO-IVUS* study [46], Cheng et al. showed a positive correlation between serum PCSK9 levels and the volume of coronary plaque necrotic core using intravascular ultrasound virtual histology (IVUS-VH) imaging, independently of LDL-C levels in a cohort of subjects at high cardiovascular risk. In the *STANISLAS* study [47], Ferreira et al. showed that high PCSK9 plasma levels were associated with arterial remodelling in a large cohort of middle-aged subjects. In addition to these findings, in our study we found that PCSK9-i treatment was able to significantly reduce PWV other than in LDL-C. Moreover, we found a positive correlation between PCSK9 levels and after treatment with PCSK9-I in FH subjects.

Overall, we can confirm that the PCSK9 levels increased in HeFH plasma after six months of high efficacy statin treatment plus ezetimibe. At the same time point, a decrease in PWV was registered. However, a further PWV reduction was observed after six months of PCSK9-i treatment, when the PCSK9 levels were significant lowered. However, we found that PWV values were associated with the PCSK9 levels at both the time points, suggesting that increased PCSK9 levels (lower delta change) could be correlated with lower PWV improvement (lower delta change). It was previously found that the PCSK9 levels increased on statin therapy [7,8,9,10,11,12,13,14,15,16,17,18]; however, there is no consensus about how this should be interpreted, in terms of efficacy of statin therapy as LLT, and in terms of CV protection. Undoubtedly, a PCSK9 increase limits the absolute magnitude of statin LDL-C lowering effect, limiting the statin-driven LDLR up-regulation (predominantly due to low intracellular cholesterol levels). Indeed, the finding that a PWV decrease was less marked in patients with a wider PCSK9 plasma level increase appears to support the hypothesis that PCSK9 could be an interesting cardiovascular biomarker of the mechanical vascular homeostasis through lipid and non-lipid pathways and it could be able to identify subjects at high ASCVD risk with a limited LDL-C lowering benefit after high intensity statin.

Finally, we can suggest that a PCSK9 increase on statin LLT could be a limitation for a full improvement of vascular function (and, of course, of LDL-C decrease), and that PCSK9 inhibition could be needed to restore the balance between PCSK9, LDL-C, LDLR, and PWV as a measure of vascular function. Indeed, this pilot study could open the way for new research on the PCSK9 role in dyslipidemias and vascular functions. Few studies had already evaluated PWV in subjects on treatment with PCSK9-i, with an added-on high intensity lipid lowering strategy. A recent study by Toth et al. showed a correlation between circulating PCSK9 concentrations and carotid intimal media thickness (cIMT) in obese and non-obese patients [48]. Another more recent study showed that PCSK9 acts on vascular smooth muscle cells, inducing cell cycle abnormalities able to contribute to the development of degenerative vascular disease [49]. Alirocumab improved artery mechanical properties in insulin-treated patients with type 2 diabetes mellitus [50]. Additionally, it was reported that PWV significantly improved after triple lipid lowering statin/ezetimibe/PCSK9-I therapy in FH patients with and without ASCVD [31,37,51].

However, no previous studies have focused on the correlation between the direct measurement of circulating PCSK9 levels and PVW, as a marker of mechanical vascular impairment, in genetically confirmed HeFH subjects. Some indirect information could be derived from interventional study with the same lipid lowering strategy and PWV measurement in this clinical setting; however, the PCSK9 levels had never been determined.

In fact, through the PCSK9 plasma levels evaluation, we found that patients experiencing a greater increase of PCSK9 levels after addition of conventional high intensity lipid lowering strategy met a less extended PWV improvement; consistently, when a strategy including a PCSK9-i was administered, the lower PCSK9 levels were associated with the wider decrease in PWV values.

There are several limitations to our study; first, the study population size was small and the timing too short to draw definitive conclusions on the role of PCSK9 as a biomarker of vascular function. A wider, prospective, randomized, controlled, ad hoc designed study is required; in this way we could better understand the potential impact of PCSK9 as a limiting factor for the vascular protective effect of conventional, high efficacy, statin-based lipid lowering therapy, and to increase the current knowledge on the desirable targets for patients at high and very high CV risk. A longer observation time and larger sample size are needed to assess the potential role of PCSK9 plasma levels on the vascular function and remodelling the effects of PCSK9-i in these pathways. Moreover, as a retrospective observation, the study was not randomized, and LLT was not assigned following a predetermined criterium. Furthermore, we restricted the observation to primary prevention HeFH subjects only.

## 5. Conclusions

In conclusion, in our study we found that the PCSK9 plasma level was correlated with PWV at baseline and its reduction was associated with a mechanical vascular improvement after PCSK9-i therapy. Further longitudinal follow-up observations are needed to evaluate the potential role of PCSK9 plasma levels on the vascular function and remodelling and the effects of PCSK9-i in these pathways.

## Figures and Tables

**Figure 1 biomolecules-12-00562-f001:**
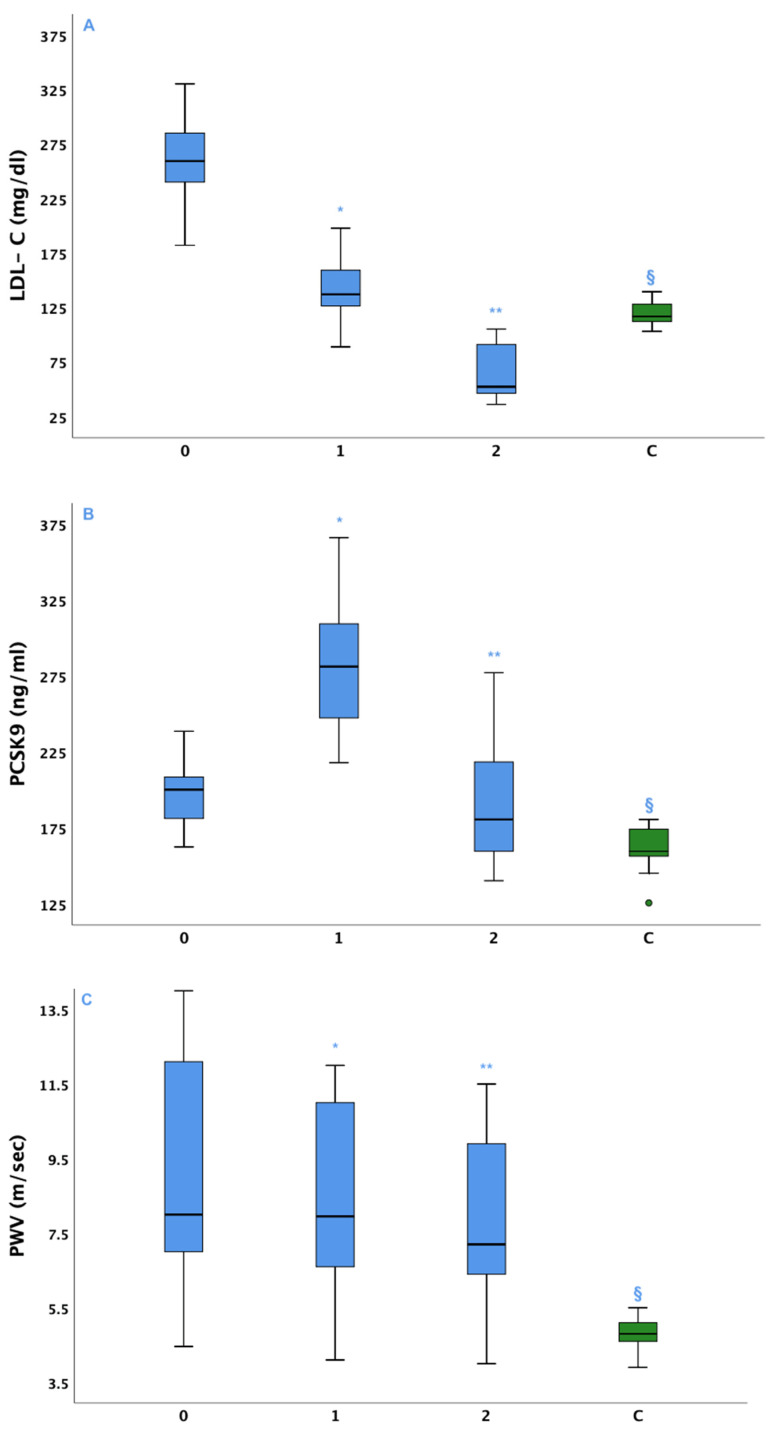
Box and whiskers plots describing LDL-C values, PCSK9 and PWV over time in HeFH and controls. Solid horizontal lines = median values; error bars = 95% Confidence intervals; Shaded area = Interquartile range. Panel (**A**): LDL-C: * *p* < 0.001 vs. baseline; ** *p* < 0.001 vs. both baseline and T1; ^§^
*p* < 0.001 vs. HeFH each timepoint. Panel (**B**): PCSK9: * *p* < 0.001 vs. baseline; ** *p* < 0.001 vs. both baseline and T1; ^§^
*p* < 0.001 vs. HeFH each timepoint. Panel (**C**): PWV: * *p* < 0.001 vs. baseline; ** *p* < 0.001 vs. both baseline and T1; ^§^
*p* < 0.001 vs. HeFH each timepoint.

**Figure 2 biomolecules-12-00562-f002:**
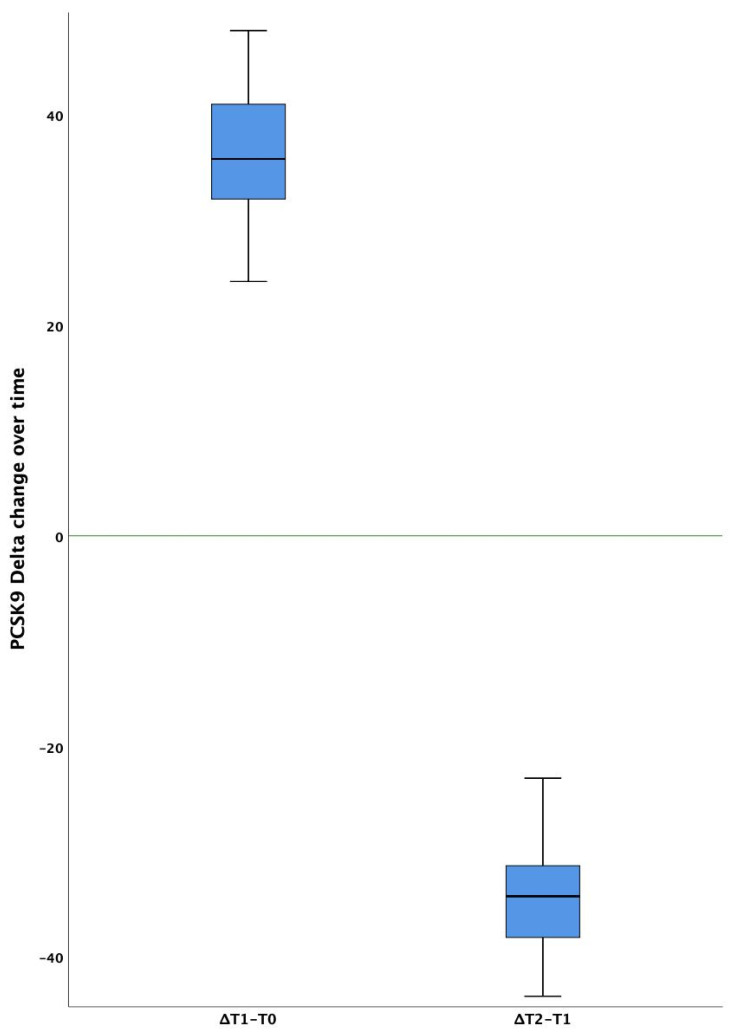
Box and whisker plots describing PCSK9 circulating levels delta change over time. Solid horizontal lines = median values; error bars = 95% Confidence intervals; Shaded area = Interquartile range. Median baseline value is depicted by the green line.

**Table 1 biomolecules-12-00562-t001:** Baseline characteristics of participants and controls.

	Controls *n* = 26	HeFH (Baseline) *n* = 26	*p*-Value
**Demographic characteristics**			
Age	43 (9)	42 (14)	0.969
Female, *n* (%)	9 (34.6)	9 (34.6)	-
BMI, kg/m^2^	23.8 (1.8)	27.1 (5.2)	0.05
ASCVD, *n* (%)	0 (0)	0 (0)	-
Hypertension	0 (0)	0 (0)	-
Type 2 Diabetes, *n*	0 (0)	0 (0)	-
Carotid plaque	0 (0)	0 (0)	-
**Lipid profile**			
TC max (mg/dL)	-	353 (49)	N/A
LDL-C max (mg/dL)	-	273 (49)	N/A
TC (mg/dL)	202 (15)	342 (48)	<0.001
HDL-C (mg/dL)	65 (7)	52 (11)	<0.001
TG (mg/dL)	90 (30)	132 (68)	0.120
LDL-C (mg/dL)	119 (11)	264 (45)	<0.001
PCSK9 (ng/mL)	161 (4)	196 (7)	<0.001
**CV risk-associated variables**			
SBP (mmHg)	121 (9)	124 (12)	0.296
DBP (mmHg)	70 (6)	77 (13)	0.068
PWV (m/s)	4.9 (0.4)	9.6 (3.1)	<0.001
**Treatment**			
Rosuvastatin 20 mg, *n* (%)	-	18 (69.2)	-
Atorvastatin 40 mg, *n* (%)	-	8 (30.8)	-
Ezetimibe 10 mg, *n* (%)	-	26 (100)	-

*p*-value: statistical significance for Mann–Whitney U test. N/A = not available.

**Table 2 biomolecules-12-00562-t002:** Lipid parameters, PCSK9 and PWV change over time in HeFH.

	T0	T1	T2	*p*-Value (T1 vs. T0)	*p*-Value (T2 vs. T1)
**Study variables**					
TC	342 (48)	212 (30)	132 (31)	<0.001	<0.001
HDL-C	52 (11)	53 (12)	48 (14)	0.75	0.35
TG	132 (68)	95 (44)	94 (29)	0.02	0.92
LDL-C	264 (45)	140 (28)	65 (26)	<0.001	<0.001
PCSK9 (ng/mL)	196 (7)	281 (8)	189 (10)	<0.001	<0.001
PCSK9 (*n*-fold)	1.22 (0.15)	1.77 (0.30)	1.17 (0.24)	<0.001	<0.001
PWV (m/s)	9.6 (3.1)	8.6 (2.5)	7.9 (2.1)	<0.001	<0.001

PCSK9 = *n*-fold difference vs. controls baseline (1 = reference value); *p*-value: statistical significance for Wilcoxon test.

## Data Availability

Data for this study are available under request.

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
