# Peer review of "PCSK9 Plasma Levels Are Associated with Mechanical Vascular Impairment in Familial Hypercholesterolemia Subjects without a History of Atherosclerotic Cardiovascular Disease: Results of Six-Month Add-On PCSK9 Inhibitor Therapy"

_biomolecules, 2022, doi:10.3390/biom12040562_

Round 1

Reviewer 1 Report

Please update the literature review in the subject area, and replan your study design to include more patients and controls

Author Response

Dear Reviewer, 

thank you for your suggestions. You can find a detailed response in the attached manuscript.

Reviewer 2 Report

To evaluate the effect of lipid lowering therapy (LLT) on circulating PCSK9 level and PWV among participants with heterozygous familial hypercholesterolemia (HeFH), this author conducted intervention study of 26 selected HeFH and 26 healthy control. From this study, this author found that after LLT was started, decreasing value of PWV was found. Statin / EZE therapy increase the plasma concentration of PCSK9 while inhibiting PCSK9 decrease the plasma levels of PCSK9. And this author also found that, there is a significant positive correlation between PWV and PCSK9 at baseline. And this author found significant positive correlation between PCSK9 plasma level change and PWV decrease. Then this author concluded that PCSK9 levels variations seemed to be correlated with PWV changes on LLT. As a reviewer, I thought this study includes newly informative knowledge. However, few issues raising in present manuscript. I commented those issues as following.

In abstract

  1. This author described as “No data exists regarding the role of PCSK9 levels in atherosclerotic injury”. And this author took PWV as a marker of atherosclerotic injury. However, PWV is an indicator of arterial stiffness but not atherosclerotic injury itself. Then not only vascular injury but also vascular repair could influence on PWV.

  1. Previous study with 56 familial hypercholesterolemia reportedthatPCSK9-i therapy significantly improved lipid and inflammatory profiles and PWV values [Ref1]. Therefore, the sentence ““No data exists regarding the role of PCSK9 levels in atherosclerotic injury” is not appropriate.

[Ref1]

Scicali R, et al. Effect of PCSK9 inhibitors on pulse wave velocity and monocyte-to-HDL-cholesterol ratio in familial hypercholesterolemia subjects: results from a single lipid-unit real-life setting. Acta Diabetol. 2021 Jul;58(7):949-957. 

  1. This author uses the abbreviation “iPCSK9” without spelling out.

  1. T1 elevates plasma concentration of PCSK9 levels while T2 decrease the level of PCSK9. And both of T1 and T2 decrease the value of PWV. However, according to description in abstract, PCSK9 plasm level change shows positively associated with PWV decrease bot in T1 (r=0.690) and T2 (r=0.602). This might mislead the reader. Using the absolute value of PCSK9 change might inappropriate.

About the study design

  1. PWV measurements can be strongly affected by blood pressure [Ref1] while levels of PCSK9 could be significantly associated with hypertension [Ref2], influence of blood pressure on present results should be taken into consideration.

[Ref1]

Yamashina A, et al. Nomogram of the relation of brachial-ankle pulse wave velocity with blood pressure. Hypertens Res. 2003;26(10):801-6. doi: 10.1291/hypres.26.801.

[Ref2]

Guo Y, et al. PCSK9: Associated with cardiac diseases and their risk factors? Arch Biochem Biophys. 2021;704:108717.

  1. How is the influence of BMI and diabetes on present results because those factors also could be associated with PCSK9 [Ref1].

[Ref1]

Levenson AE, et al. Obesity and type 2 diabetes are associated with elevated PCSK9 levels in young women. Pediatr Diabetes. 2017;18(8):755-760.

In discussion section

  1. From present results, I could not understand the clinical perspective of evaluating plasma concentration of PCSK9. If this author thought measuring PCSK9 value is efficient tool to evaluating the effect of LLT, this author should show the effectiveness of measuring PCSK9 than that of measuring the lipid values. Then at least lipid adjusted model should be taken into consideration. And the clinical implication for present study should be described in discussion section.

  1. Even there are many similar studies that reported the influence of PCSK9 inhibitor on PWV [Ref1-Ref3], only limited studies were described in present manuscript. Furthermore, there is no description that explain what is the different from previous similar studies. This can be strength the present study.

[Ref1]

Scicali R, et al. Effect of PCSK9 inhibitors on pulse wave velocity and monocyte-to-HDL-cholesterol ratio in familial hypercholesterolemia subjects: results from a single lipid-unit real-life setting. Acta Diabetol. 2021 Jul;58(7):949-957. 

[Ref2]

Scicali R, et al. Analysis of arterial stiffness and sexual function after adding on PCSK9 inhibitor treatment in male patients with familial hypercholesterolemia: A Single Lipid Center Real-word Experiences. J Clin Med. 2020 ;9(11):3597. 

[Ref3]

Mandraffino G, et al. Arterial stiffness improvement after adding on PCSK9 inhibitors or ezetimibe to high-intensity statins in patients with familial hypercholesteromia: A two-lipid center real-word experience. J Clin Lipidol. 2020;14(2):231-240. 

Author Response

Dear Reviewer, 

thank you for your suggestions. You can find a detailed response in the attached file.

Reviewer 3 Report

Authors should address the following concerns before acceptance:

  1. Please cite appropriate reference for measuring pulse wave velocity to showcase it can be a surrogate marker of early vascular injury
  2. In Table 2, authors should explain the declining of HDL-C level
  3. Please provide a box-whisker plot for PCSK9 level to understand the variation
  4. Is the inhibitor PCSK9 specific? author should discuss. Please modify the images/tables indicating the PCSK inhibitor starting time.
  5. The final results should be adjusted (with BMI) when comparing with controls (as done by showing fold-changes).
  6. Do the authors have follow-up levels of controls at different time point?

Author Response

(The authors gave the same response as above.)

Round 2

Reviewer 1 Report

Minor linguistic editing is needed, after proofreading by a native speaker or an English language expert.

Author Response

We thank the Reviewer for the suggestion; accordingly, a revision of the english style and language has been performed.

Reviewer 2 Report

Thanks to this author’s great effort, I thought this manuscript improved well. However, I still found one issue that should be mentioned.

Even this author focused on the atherosclerotic injury, this author used PWV that indicates the functional value of arterial stiffness. Then present expression “atherosclerotic injury” is not correct. It could mislead. If this author wants to evaluate the atherosclerotic injury, this author should change present study plan.

Author Response

We thank the Reviewer for the suggestion; accordingly, the term "atherosclerotic injury has been changed to "mechanical vascular impairment" in all manuscript sections.